# Impact of Early Hemoglobin Levels on Neurodevelopment Outcomes of Two-Year-Olds in Very Preterm Children

**DOI:** 10.3390/children10020209

**Published:** 2023-01-25

**Authors:** Catherine Gire, Ninon Fournier, Johanna Pirrello, Stéphane Marret, Hugues Patural, Cyril Flamant, Véronique Pierrat, Monique Kaminski, Pierre-Yves Ancel, Barthélémy Tosello, Julie Berbis

**Affiliations:** 1Department of Neonatology, North Hospital, APHM University Hospital, Chemin des Bourrely, 13015 Marseille, France; 2EA3279, Self-Perceived Health Assessment Research Unit, Faculty of Medicine, Aix Marseille University, 13385 Marseille, France; 3Department of Neonatal Pediatrics, Intensive Care, and Neuropediatrics, Rouen University Hospital, 76000 Rouen, France; 4INSERM Unit 1245, Team Perinatal Handicap, School of Medicine of Rouen, Normandy University, 14000 Normandy, France; 5Department of Neonatal Medicine, Saint-Etienne University Hospital, 42100 Saint-Etienne, France; 6Department of Neonatal Medicine, Nantes University Hospital, 44093 Nantes, France; 7Department of Neonatal Medicine, CHU Lille, Jeanne de Flandre Hospital, 59000 Lille, France; 8CRESS, Obstetrical Perinatal and Pediatric Epidemiology Research Team, EPOPé, INSERM, INRAE, Université Paris Cité, 75004 Paris, France; 9Faculté de Médecine de Marseille, Aix Marseille University, CNRS, EFS, ADES, 13385 Marseille, France

**Keywords:** hemoglobin, very preterm, 2-year neurodevelopmental outcomes

## Abstract

Objective: To evaluate, in very preterm infants, the hemoglobin (Hb) levels during the first 24 h and the neurodevelopment outcomes at 24 months of corrected age. Design, setting, and patients: We conducted a secondary analysis of the French national prospective and population-based cohort EPIPAGE-2. The eligible study participants were live-born singletons who were born before 32 weeks of gestational age, with early Hb levels who were admitted to the neonatal intensive care unit. Main outcome measures: The early Hb levels for an outcome survival at 24 months of corrected age without neurodevelopmental impairment were measured. The secondary outcomes were survival at discharge and without severe neonatal morbidity. Results: Of the 2158 singletons of <32 weeks with mean early Hb levels of 15.4 (±2.4) g/dL, 1490 of the infants (69%) had a follow-up at two years of age. An early Hb of 15.2 g/dL is the minimum receiving operating characteristic curve at the 24 months risk-free level, but the area under the curve at 0.54 (close to 50%) indicates that this rate was not informative. In logistic regression, no association was found between early Hb levels and outcomes at two years of age (aOR 0.966; 95% CI [0.775–1.204]; *p* = 0.758) but rather there was a correlation found with severe morbidity (aOR 1.322; 95% CI [1.003–1.743]; *p* = 0.048). A risk stratification tree showed that male newborns of >26 weeks with Hb of <15.5 g/dL (n = 703) were associated with a poor outcome at 24 months (OR 1.9; CI: [1.5–2.4] *p* < 0.01). Conclusions: Early low Hb levels are associated with major neonatal morbidities in VP singletons, but not with neurodevelopment outcomes at two years of age, except in male infants of >26 Weeks GA.

## 1. Introduction

Except for extremely preterm infants, Hb levels are higher at birth than later in life in order to compensate for relative arterial hypoxia. After birth, the Hb level drops since the red blood cell (RBC) death rate is faster than its replacement rate. A physiological anemia occurs in all new-born infants in the first months, which is exacerbated in very preterm infants (VP) [1]. An incidence of anemia is partially linked to a defect in postnatal renal erythropoietin synthesis and to blood samples that are taken during the initial hospitalization [2]. An RBC transfusion remains the leading therapy. Approximately 50% of children who are born under 32 weeks of GA, and more than 85% of children under 1000 g, benefit from an RBC transfusion during their neonatal care hospitalization [3]. However, in VPs, RBC transfusion could be associated with severe bronchopulmonary dysplasia (BPD), necrotizing enterocolitis (NEC), high-grade intraventricular-hemorrhage (IVH), or retinopathy of prematurity (ROP) [4]. 

RBC transfusions are currently obtained from adult donors. Neonatal RBCs contain mainly fetal hemoglobin (FHb), which binds with oxygen with a greater affinity than adult hemoglobin (HbA) [5]. The fraction of FHb is approximately 70 to 80% during the third trimester of pregnancy, with a gradual decrease to 3–4% at six months after term age [6]. It has been hypothesized that FHb might act as a protection against oxidative stress. A proposed underlying mechanism involves a shift in the oxygen dissociation curve, due to the replacement of FHb by HbA, with a consequentially higher fraction of oxygen dissolved in the plasma and increased tissue exposure to oxygen [7,8].

The recommendations to prevent VPs’ RBC transfusions include delayed cord clamping (DCC), erythropoietin, and decreasing the frequency of the taking of blood samples. This practice remains limited, despite noteworthy, supportive scientific evidence [9,10,11]. 

In VPs, the early Hb levels may be improved by DCC with an additional blood flow after childbirth and increasing the Hb levels for several minutes, or even hours, after birth. The additional placental blood flow by DCC for 30 to 120 s appears to be associated with better circulatory stability, a lower rate of respiratory distress syndrome, fewer blood transfusions, and a lower risk of IVH and NEC [12]. Thus, the Hb threshold from the cord blood could be a direct indicator of blood volume and the child’s pre-birth basal state. A recent study showed that DCC for 45 s increases the Hb level at birth in comparison to immediate cord clamping (15.9 g/dL vs. 14.9 g/dL, *p* = 0.001), reducing the need for RBC transfusions in very-low-birth-weight infants (VLBW) [13]. A single multicenter observational study showed that a low, early Hb level is an independent mortality risk factor that increases the likelihood of a VPs’ RBC transfusion, regardless of the delivery mode and DCC [14].

Few data exist regarding minimal Hb levels at birth “without risk” of short-term morbidity, long-term morbidity, and mortality with or without DCC. Further studies are needed in order to correlate early Hb levels at birth with long-term neurodevelopmental outcomes.

We hypothesized that a higher Hb threshold in the first 24 h after birth improves the prognosis at 24 months. Our objective was to evaluate the VP singletons’ minimal value of early Hb to an optimal neurodevelopment outcome at 24 months in the EPIPAGE-2 cohort [15]. 

## 2. Method

### 2.1. Study Design and Population 

This study is a secondary analysis of the French national population-based and prospective cohort EPIPAGE-2 (Etude éPIdémiologique sur les Petits Ages GEstationnels-2) [16,17]. Our study included infant singletons who were admitted alive to the neonatal intensive care units (NICUs) and who were born before 32 weeks of GA and without congenital brain malformation or chromosomal abnormalities. The infants with missing early Hb levels during their first 24 h of life were not included. 

### 2.2. Measurements Collected

The main variables of interest were as follows: early Hb values; lowest Hb rate during NICU hospitalization and Hb rate at discharge; RBC transfusions and their number; transfusion sparing; and DCC and treatment with erythropoietin. Severe neonatal morbidity was defined by at least one of the following: high-grade IVH and/or periventricular cystic leukomalacia; NEC grade 2–3; ROP ≥ stage 3; severe BPD; and/or a need for oxygen therapy for a minimum of 28 days after birth plus the need for 30% or more oxygen and/or mechanical ventilation and/or continuous positive airway pressure at least until 36 weeks of corrected age [16].

#### Outcomes

The primary outcome was survival occurrence without neurosensory impairment or the occurrence of a severe sensorimotor disorder and/or death at 24 months. The secondary outcomes were death during the neonatal stay or survival at discharge without severe morbidity. Survival at 24 months without neurosensory impairment was defined as no cerebral palsy and/or without any neurodevelopment disorders. Cerebral palsy was defined according to the diagnostic criteria of the Surveillance of Cerebral Palsy in Europe network (SCPE) [17]. Neurodevelopment was assessed using the Age and Stages Questionnaire (ASQ) at 24 months of age. A neurodevelopment disorder was defined as an ASQ score of less than two standard deviations (-2SD) in one of the five assessed domains [18].

### 2.3. Ethics

The National Commission for Informatics and Liberties (CNIL no.911009), the Advisory Committee on the Processing of Information in Research Matters (Reference no. 10.626), and the Committee of Protection of Persons participating in biomedical research (Reference CPP SC-2873) approved this study.

### 2.4. Statistical Analysis

Numbers and percentages (%) described the categorical variable data, and the quantitative data was shown in means and standard deviations (SD) or the median and interquartile range. The results were expressed as odds ratio (OR) or adjusted odds ratio (aOR) with a 95% confidence interval (95% CI). A *p*-value < 0.05 was considered statistically significant. To account for the duration of the recruitment periods by GA at birth, a weighted coefficient was allocated to each individual (1 for births at 22–26 weeks, 1.346 for births at 27–31 weeks, and 7 for births at 32–34 weeks). 

A comparative analysis was carried out between the perinatal data of the children who were followed-up at 24 months and those who were not.

To determine whether an early low Hb level was without risk to neurodevelopment at 24 months, we searched for a minimum risk-free threshold using a receiving operating characteristic (ROC) curve and then associations with the transfusion thresholds (10–12 g/dL) during the first weeks after birth, which was already studied in two large RCT clinical trials. These two large studies, ETTNO and Top [19,20], showed that Hb transfusion values within the range of 11–13.7 g/dL for critically ill infants, and 9–12 g/dL, for stable infants, can be used safely, without negative survival consequences or adverse neurodevelopmental consequences [1]. The sensitivity, specificity, and positive and negative predictive value of the early Hb level for 24 months without sensory neurodevelopment impairment was determined by a ROC curve.

The association between early Hb, ROC Hb, Hb thresholds, and composite endpoints at 24 months, and in the neonatal period, was achieved in a multivariate analysis with adjustment for antenatal corticosteroid therapy, GA, birth weight, sex, Apgar score, transfusion, erythropoietin treatment, DCC, severe neonatal morbidity, and socio-economic conditions. Regression logistic analyses were conducted with methods allowing data imputation for missing information. 

Finally, by taking the negative outcomes at 24 months, it was possible to study the influence of explanatory variables such as GA, sex, birth weight, neonatal morbidity, ROC Hb thresholds, and transfusions. Using a recursive partitioning analysis method, we built a classification tree by dividing the population into either normal or abnormal neurodevelopment at 24 months. This algorithm established a risk stratification tree for non-optimal development.

## 3. Results 

### 3.1. Population

This study’s population characteristics are detailed in Figure 1 and Table 1.

This study included 2158 singleton births with a mean of 28.7 weeks (±1.997) and a mean of 1201 (±361) g birth weight (Figure 1). The early mean Hb (SD) level was 15.4 (±2.4) g/dL, with increases depending on the GA. The RBC transfusion rate was 42% (n = 956), with an average of 2.1 (±1.65). Concerning the comparison of the perinatal data between the infants who were followed up at 24 months and those who were not, the two groups differed significantly depending on the mother’s age, parity, and socio-economic conditions (Appendix A).

### 3.2. Values of the Hb Threshold

The Hb level at 15.2 g/dl appeared to be the most discriminating value for an outcome without neurosensory impairment at two years of age (AUC: area under the curve) as follows: OR 0.540; 95% CI [0.513–0.568]; *p* = 0.005 (Appendix A). The positive predictive value (PPV) and the negative predictive value (NPV) remained very modest (respectively 65.1% and 40.7%) (Appendix A). 

According to the ROC Hb threshold of 15.2 g/dL, 47% of the children had an Hb level of <15.2 g/dL at 24 h after delivery (Appendix A). The two populations varied significantly according to weeks GA, BW, morbidity, mortality, transfusion, and cerebral palsy at 24 months. Survival without neurosensory impairment was different, but not significant. 

The Hb threshold of <15.2 g/dL was correlated with morbidity (aOR 1.322; 95% CI [1.003–1.743]; *p* = 0.048) but not with mortality (aOR 1.009; 95% CI [0.703–1.448]; *p* = 0.962) in the multivariate analysis (Table 2). 

### 3.3. Neurodevelopmental Outcomes at 24 Months Corrected Age

There was no significant independent association between the early ROC Hb threshold of <15.2 g/dL and the outcome at 24 months (aOR 0.966; 95% CI [0.775–1.204]; *p* = 0.758) (Table 3). The results remained unchanged with the Hb threshold of 10 and 12 g/dL, with a continued Hb value, and by subgroup of GA (Appendix A).

The reference group with the most favorable outcomes (i.e., no fatalities or neurosensory impairment) was Group 4, with 316 female patients (18.3%) with a GA of >26 +3 weeks and early Hb levels of >15.2 g/dL. The level of risk of unfavorable outcomes, as compared to group 4, was Group 1, with an extremely preterm GA of <26 + 3 weeks (OR = 3.5 [2.6; 4.7]; *p* < 0.001), n = 390 (22%); Group 2, with male infants of ≥26 + 3 weeks and an Hb level of <15.2 g/dL (OR = 1.9 [1.5; 2.4]; *p* < 0.01), n = 702 (41%); and Group 3, with female infants and an Hb level of <15.2 g/dL (OR = 1.3 [1.0; 1.8]; *p* = 0.03) n = 312 (19%) (Figure 2).

## 4. Discussion 

We found no significant correlation between the early Hb threshold of <15.2 g/dL and the outcome at 24 months, except for the non-extremely preterm boys and perinatal death. 

The early neonatal Hb transfusion thresholds (restrictive or liberal) within the range of 9–13 g/dL can be used in VPs during the first weeks after birth without expecting any adverse consequences for survival or neurodevelopment [21,22]. These findings are consistent with the failure of erythropoietin to improve neurodevelopmental outcomes in VP infants despite the increase in RBC precursors, as with our results [23]. Additionally, an early Hb level, as with the other blood lines, is very dependent on weeks and increases with weeks [24,25]. This explains the results of our decision tree, i.e., an early Hb threshold of >15.2 would have no independent impact at 24 months on a population that is said to be less at risk (female, VP, >26 weeks), which naturally has a higher sex-related Hb fetal threshold [26]. A recent study confirms this sex-specific association between an early marker of anemia and cognitive function at age 12 months of age and white matter volume [27]. 

Higher early Hb levels may lead to better hemodynamic stability, which may reduce morbidity, as shown in our study’s correlation between the Hb levels that are less than 15.2 g/dL and severe morbidity [28,29,30]. A meta-analysis comprising seven randomized control trials (RCT) of 501 patients on umbilical cord milking, resulting in higher Hb levels from birth, demonstrated a reduced risk of severe BPD and IVH of all of the grades [30]. A multicenter observational study of 890 VPs confirmed a relationship between an early low Hb level of <15 g/dL and mortality, without any outcome association [14]. Other studies have revealed the following: i.a prolonged ventilation in cases of early Hb levels of <15 g/dL at birth; VPs with higher early Hb levels have a reduced need for respiratory support and intensive care; and a low fraction of FHb during the first week of postnatal life is independently associated with the development of retinopathy in VP infants [13,31,32].

However, DCC, with A level evidence, was performed on only 4% of our EPIPAGE-2 population in 2011 [9]. A meta-analysis showed an association with an increased Hct at four hours after birth, which lessens the early transfusions and IVH [29]. Thus, a low early Hb level is inherently associated with increased RBC transfusions in VLBW individuals; however, in those who were small for their gestational age with severe anemia of <8 g/dL, an independence of RBC transfusions is associated with an increased risk of NEC [31]. In our study, 42% of the children were transfused more often when they had less than 15 g/dL Hb. RBC transfusion is the most widely used strategy to correct anemia, but it is not without its risks.

A multicenter observational study with 1077 preterm individuals (23–36.9 weeks GA; PN: 400–1495 g) showed that death is more prominent among children who receive an RBC transfusion within the first 28 days after delivery. Transfusions are linked to ROP, BPD, and IVH [32,33,34,35,36]. The hypothesis is that these morbidities result from a non-physiological release of oxygen from the adult Hb in immature tissues [32].

Unlike the other published studies, our sample comes from a prospective cohort of EPIPAGE-2, which is based on a national level population (93% of VPs born in France in 2011), thus reflecting the diversity of French neonatal care practices. Our study’s main limitation is the proportion of missing data that is related to the loss of follow-up at 24 months. The attrition was 81%, and was deemed to be good relative to the cohort size and its geographic scope. The comparison of the perinatal data between the follow-ups and the non-follow-ups indicated that the children were comparable, except for the socio-economic status. Even if this is a large-scale, prospective, multicenter study confirming the hypotheses of the RCT on DCC studies that have been published previously, it is difficult to establish a chronological cause and effect link between the early Hb levels, the neonatal complications, and the 24-month outcomes. Although we adjusted for confounding factors, such as transfusions and those issues usually impacting the long-term outcomes (such as weeks, weight, sex, and socio-economic conditions), other factors, such as treatments influencing the prognosis, were not considered. Additionally, we hypothesize that a subtle neurodevelopmental abnormality may not be apparent at 24 months of age, but might manifest afterwards. Therefore, each domain cannot be considered only separately; impairment in one skill can impact the development of the child’s other abilities, and neurodevelopment needs to be assessed from a holistic approach during their infancy [37]

Finally, this study suggests that early low Hb levels (<15 g/L) represent a risk factor for severe neonatal morbidity in VP populations. Similarly, a high level of FHb is not necessarily neuroprotective in the long term, especially in males. The limited data that is currently available suggest that sex influences the long-term consequences of the transfusion thresholds. Indeed, males do not tolerate restrictive transfusion policies, therefore, the liberal transfusion policy may be appropriate for females [38].

The FHb may affect fractional oxygen extraction from peripheral muscles, but it does not appear to affect cerebral oxygenation in preterm infants, as it is sex-linked [38]. Further studies are needed in order to draw a definitive conclusion, especially with regard to the changes in oxygenation that are induced by transfusions of adult RBC due to sex differences in ferritin and in inflammatory response [38,39]. 

We assessed the infants until they were 24 months old on their neurodevelopment, but we have not yet analyzed the data on the longer-term prognosis in childhood. A comparison at five years on the same EPIPAGE-2 cohort regarding the sex-dependent, early Hb, and long-term neonatal outcomes, regardless of the number of transfusions, would be ideal.

## Figures and Tables

**Figure 1 children-10-00209-f001:**
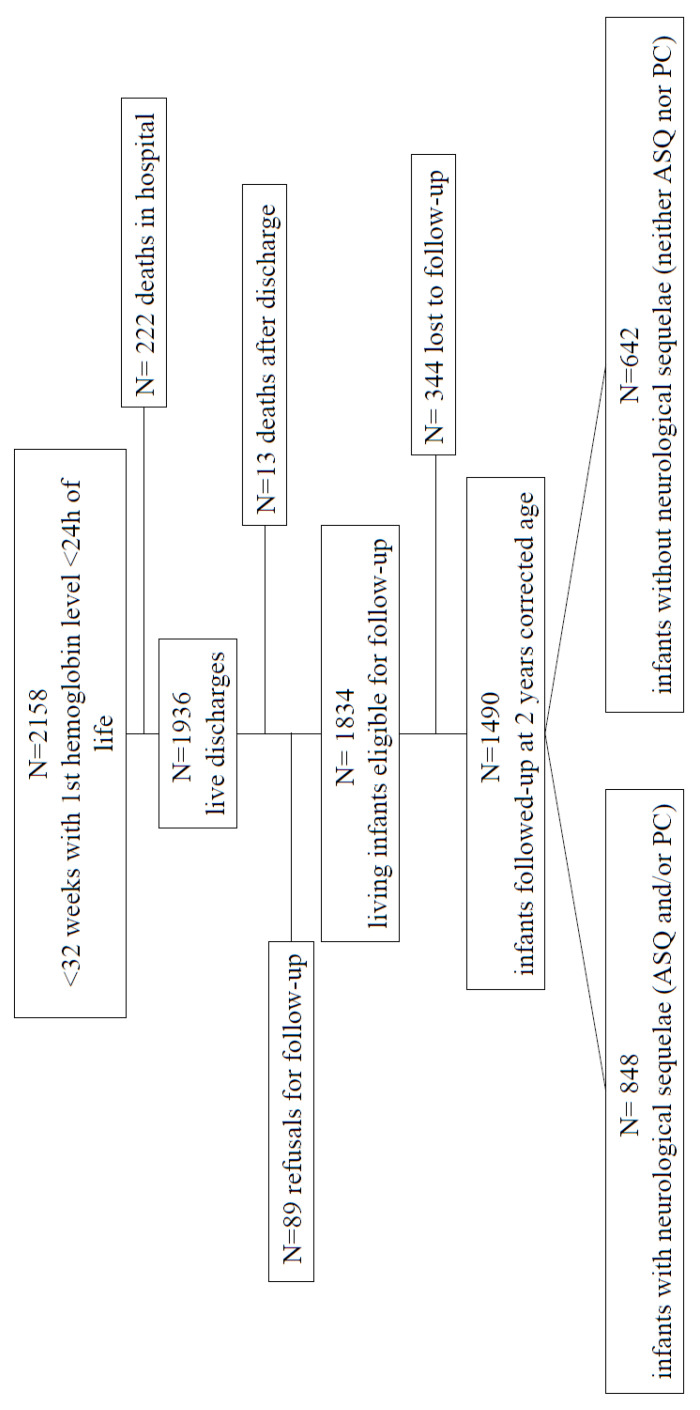
Study population’s flow chart.

**Figure 2 children-10-00209-f002:**
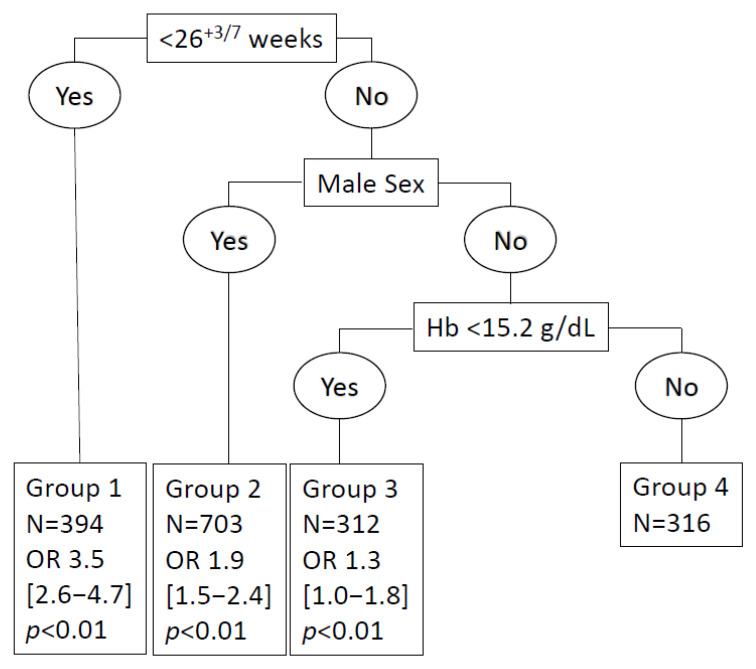
Risk stratification tree to predict non-optimal neurodevelopmental outcomes at 24 months.

**Table 1 children-10-00209-t001:** Maternal, obstetrical, and neonatal characteristics.

	Population, n = 2158	Missing Data
**Maternal Characteristics**		
Age at childbirth in years (mean ± SD)	29.4 (±6.0)	1 (0.05)
Parity (mean ± SD)	1.58 (±1.87)	3 (0.1)
Multiparity	1389 (64.3)	
*Socio-economic status*		126 (5.8)
Higher and intermediate professions	1329 (61.9)	
Other	611 (28.1)	
No occupation	92 (4.2)	
Tobacco use during pregnancy	525 (24.4)	80 (3.7)
Antidiabetic treatment	78 (3.7)	71 (3.3)
**Cause of prematurity**		
Idiopathic premature labor	826 (37.3)	74 (3.5)
Premature rupture of membranes	496 (22.7)	74 (3.5)
Vascular placenta pathology	585 (27.9)	74 (3.5)
Without IUGR	285 (13.6)	
With IUGR	300 (14.3)	
Isolated IUGR	113 (5.5)	
Isolated retro-placental hematoma	64 (3.1)	
Metrorrhagia	97 (4.5)	11 (0.5)
Maternal infections	494 (22.2)	395 (18.2)
Other causes of premature birth *	138 (6.5)	11 (0.5)
**Delivery**		
Gestational age (GA) (mean ± SD)	28.7 (±1.997)	0
≤27 Weeks GA	678 (27.4)	
(28–32) Weeks GA	1480 (72.6)	
Cesarean	355 (63.9)	25 (1.2)
Antenatal corticosteroid therapy	1713 (79.6)	45 (2.1)
Antenatal magnesium sulfate	195 (9.2)	36 (1.6)
Delayed cord clamping	87 (4.1)	96 (4.4)
**Newborn**		
Birth weight, g (mean ± SD)	1201(±361)	0
≤ 1000 g	785 (33)	
>1000 g	1373 (67)	
Male sex	1129 (52.3)	370 (16.7)
Apgar at 10 min (mean ± SD)	9.1 (±1.5)	
Apgar <7 at 10 min	106 (4.7)	370 (16.7)
Tracheal intubation	1266 (56.7)	51 (2.4)
Oxygen therapy	1552 (71.4)	94 (4.3)
Chest compressions	116 (5.2)	86 (3.9)
**Hb, transfusion, Erythropoietin**		
Hb at birth, g/dL (mean ± SD)	15.4 (±2.4)	0
Centiles 25 Centile 50 Centiles 75	13,8462 15,4000 16,8000	
Hb at birth GA		
22–25^+6^, (n = 216)	13,8 (±2.14)	
26–27^+6^, (n = 536)	14,6 (±2.24)	
28–31^+6^, (n = 1992)	15,7 (±2.32)	
Lowest Hb level during hospitalization, g/dL (mean ± SD)	10.3 (±3.4)	107 (5)
Hb at discharge, g/L (mean ± SD)	11 (±1.97)	500 (22.6)
Transfusion	956 (42)	27 (1.3)
Number of RBC transfusions (mean ± SD)	2.1 (±1.65)	12 (1.2)
Without erythropoietin	1101 (51.6)	21 (0.9)
**Neonatal morbidity**		
Neonatal morbidity **	397 (17.2)	157 (7)
Severe BPD	131 (5.6)	314 (13.3)
Stage 2–3 NEC	80 (3.6)	42 (1.8)
ROP stage 3–4	17 (0.6)	1348 (63)
Severe brain abnormalities	209 (9)	38 (1.7)
Death (before discharge)	222 (9.2)	0

Data are presented in percentages (%), unless otherwise indicated, and % are weighted according to GA. Abb: IUGR: intrauterine growth retardation; SD: standard deviations; weeks of GA (gestational age); Hb: hemoglobin; RBC: red blood cells. * Other causes of prematurity including isolated and sporadic explications (acute hepatic steatosis of pregnancy, severe fetal anemia, sickle cell anemia, psychiatric reasons, etc.). ** Neonatal morbidity was defined by the following: severe bronchopulmonary dysplasia (BPD), severe brain abnormalities (severe periventricular cystic leukomalacia or severe intraventricular hemorrhage (IVH) grade III or IV), necrotizing enteritis colitis (NEC) stage 2–3, or severe retinopathy of prematurity (ROP) > stage 3.

**Table 2 children-10-00209-t002:** Perinatal outcome comparisons at birth depending on the Hb threshold (15.2 g/dL).

	Hb > 15.2 g/dL n (%)	Hb <= 15.2 g/dL n (%)	*p*-Value	aOR after Multivariate Analysis (CI95%)	*p*-Value
Death	89/1181 (6.9)	133/977 (12.1)	<10^–3^	1.009 (0.703–1.448)	0.96
Neonatal morbidity *	184/1104 (15.8)	213/897 (21.9)	<10^–3^	1.322 (1.003–1.743)	0.04

Percentages and *p* values are weighted according to GA. * Neonatal morbidity was defined by the following: severe bronchopulmonary dysplasia (BPD), severe brain abnormalities, severe periventricular cystic leukomalacia or severe intraventricular hemorrhage (IVH grade III or IV), ulcerative necrotizing enterocolitis (NEC) stage 2–3, or severe retinopathy of prematurity (ROP) > stage 3. SD: standard deviations; Hb: hemoglobin; aOR: adjusted odds ratio. Multivariate analysis: odds ratio adjusted for GA, antenatal corticosteroid therapy, birth weight, sex, Apgar score, neomorbidity, socio-economic status for mortality, for GA, antenatal corticosteroid therapy, birth weight, sex, Apgar score, and socio-economic status for neonatal morbidity.

**Table 3 children-10-00209-t003:** Correlation between the hemoglobin value at birth (hemoglobin threshold at 15.2 g/dL) and 2-year prognostics for infants born <32 weeks.

	Patients N/Total (%)	aOR after Multivariate Analysis	*p*-Value	aOR after Multivariate Analysis and Imputation	*p*-Value
		(IC95%)		(IC95%)	
**2-year Prognosis**					
Cerebral palsy	80/1588 (5)	1.121 (0.646–1.944)	0.685	1.129 (0.639–1.996)	0.675
High risk of developmental delay *	835/1512 (55.2)	0.974 (0.781–1.214)	0.812	0.949 (0.760–1.186)	0.646
High risk of developmental delay and/or PC	848/1490 (57)	0.961 (0.768–1.203)	0.731	0.960 (0.769–1.200)	0.723
Alive without sequelae **	642/1725 (38.1)	1.039 (0.833–1.295)	0.737	0.966 (0.775–1.204)	0.758
Death or disability ***	1083/1725 (61.9)				

aOR: Adjusted odds ratio; 95%; CI: 95% Confidence interval; PC: Cerebral palsy; ASQ: Ages and Stages Questionnaire odds ratio adjusted for gestational age, antenatal corticosteroid therapy, birth weight, sex, Apgar score, neomorbidity, socio-economic status, and transfusion factor. * ASQ below threshold. ** Living without pathological ASQ or cerebral palsy. *** Death or living with pathological ASQ and/or cerebral palsy.

## Data Availability

The datasets that were generated and/or analyzed during the current study are not publicly available as the data belongs to the Assistance Publique Hopitaux de Marseille. However, the datasets are available from the corresponding author on reasonable request and after signing a contract pertaining to the provision of data and/or results.

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
