# Peer review of "Impact of Early Hemoglobin Levels on Neurodevelopment Outcomes of Two-Year-Olds in Very Preterm Children"

_children, 2023, doi:10.3390/children10020209_

Round 1
Reviewer 1 Report
This is a prospective population-based cohort study. The study included infant singletons admitted alive to the Neonatal Intensive Care Units (NICUs) and who were born before 32 weeks gestational age (GA). The authors conclude that early low hemoglobin (Hb) levels are associated with major neonatal morbidities in very preterm singletons, but not with neurodevelopment outcomes at two years of age.
1.The low Hb (Hb<15.2g/dL) group infants have lower birth weight, GA and they have lower rate of antenatal corticosteroid, delayed cord clamping, lower average Apgar at 10 minutes. They also have higher rate of oxygen therapy and tracheal intubation than high Hb (Hb>15.2g/dL) group infants. All above factors may contribute to major neonatal morbidities. Early low Hb levels are associated with above factors that may contribute to major neonatal morbidity in very preterm singletons. It is difficult to conclude that low Hb levels during the first 24 hours after birth have relationship with major neonatal morbidity in very preterm singletons.
2.Early low Hb levels are not associate with neurodevelopment outcomes at 2 years of age in this study. This finding is lack of well discussion in this manuscript. Many factors may affect neurodevelopment outcomes, not low Hb levels only.
Author Response
Authors’ Response to Reviewers:
Dear editor and reviewers,
Thank you for your letter and for the reviewers’ comments on our manuscript entitled “Impact of early hemoglobin levels on neurodevelopment outcomes of two-year old in very preterm children.”
All of these comments were very helpful for revising and improving our paper. We have studied these comments carefully and have made corresponding corrections that we hope will meet with your approval. The changes in the revised manuscript are marked in red. The responses to the reviewers’ comments are provided below.
We would like to express our great appreciation to you and the reviewers for the comments on our paper.
Kind regards,
- Reviewer 1:
Dear Doctor, Dear Reviewer,
We thank you so much for your valuable and appreciated inputs in our article. The interest you showed in this regard encourages us to continue our research.
Respectfully
Reponses:
Your first observation relates to Table 3 Supp where indeed some perinatal and neonatal data are significantly different between the two groups depending on the Hb threshold. It is established in the literature that the lower the threshold the greater the neonatal morbidity. In our study, this was not the purpose, but rather to establish a minimum Hb level from which complications are less and the neurological outcome is also less.
Finally, in response to comment #2, Table 5 Supp confirms that this threshold is decisive for neonatal morbidity but not influential for neurodevelopmental outcome.
Reviewer 2 Report
dear authors,
I've appreciated your paper the topic is nice and interesting, the study well conducted and well written.
neurodevelopment outcome of very preterm baby is major issue so the implementation of studies to better understand potential underlying associated with adverse outcomes is crucial.
I would just add a table with a summary of findings
Author Response
Dear Doctor, Dear Reviewer,
We thank you so much for your valuable and appreciated inputs in our article. The interest you showed in this regard encourages us to continue our research.
Your request is legitimate but the instructions to the authors require us not to exceed a certain number of tables, moreover a summary table would be redundant with the results and not accepted by the editorial committee. Thank you for your understanding.
Respectfully